# High-Temperature Oxidation Behavior of Fe–1Cr–0.2Si Steel

**DOI:** 10.3390/ma13030509

**Published:** 2020-01-21

**Authors:** Mingxin Hao, Bin Sun, Hao Wang

**Affiliations:** 1School of Mechanical Engineering, Shenyang University, 21 Wanghua South Street, Shenyang 110044, China; haomingxin@foxmail.com; 2State key laboratory of Rolling and Automation, Northeastern University, 11 Wenhua Road, Shenyang 110819, China; whaofom@foxmail.com

**Keywords:** high-temperature oxidation, oxide scale, weight gain, activation energy for oxidation, Cr oxide

## Abstract

In the case of Fe–1Cr–0.2Si steel, tube furnace oxidation was carried out for 120 min and 30 min. These studies, along with the high-temperature oxidation behavior of Fe–1Cr–0.2Si steel, were examined from 700 to 1100 °C. It has been observed that with an increase in the oxidation time, the oxidation weight gain per unit area of Fe–1Cr–0.2Si steel changed from a linear to a parabolic relationship. The time was shortened when the oxidation phase was linear. When the oxidation temperature exceeded 900 °C, the value of *W_Transition_* decreased, and the oxidation rule changed. It could be considered that overall, the iron oxide structure of Fe–1Cr–0.2Si steel is divided into two layers. The formation of an outer oxide of iron is mainly caused by the outward diffusion of cation, while the inward diffusion of O ion forms the inner oxides of chromium and silicon. As the temperature increases, the thickness of the outer iron oxide gradually increases, and the thickness ratio of the inner mixed layers of chromium- and silicon-rich oxides decreases; however, the degree of enrichment of Cr and Si in the mixed layer increases. After high-temperature oxidation, Cr and Si did not form a composite oxide but were mechanically mixed in the form of FeCr_2_O_4_ and Fe_2_SiO_4_, and no significant delamination occurred.

## 1. Introduction

Adding an alloy consisting of Cr and Si elements to low-carbon steel can improve its heat resistance and environmental corrosion resistance. Therefore, Fe–Cr and Fe–Si alloys are widely used in the fields of heat-resistant alloy and stainless steel [1,2]. Studies have shown that when a specific amount of Si was added to low-carbon steel, after the oxidation of Si during heating, it first enriches at the interface between the scale and iron matrix to form SiO_2_ by a chemical reaction with oxygen, following which a series of complex chemical reactions take place with iron ions to finally form Fe_2_SiO_4_. The above-formed oxide reduced the rate and amount of diffusion of Fe to the surface of the steel substrate in the subsequent heating process, and also led to the failure of oxygen to react with iron ions smoothly, thereby improving the oxidation resistance of the oxide sheet [3,4].

After adding Cr to the iron-based alloy, the product formed by oxidation is affected by the amount of added Cr [5,6,7]. With the addition of 5% Cr, the Cr-rich and Fe oxides enter the FeO phase to form FeCr_2_O_4_, but the solubility of the spinel structure is also limited owing to the stability of the spinel structure. At a high concentration of FeO defects, the concentration of vacancies was not significantly detected [8], and hence there is no increase in the oxidation rate constant. As the content of Cr increases, the Fe^2+^ ions are surrounded by island-like FeCr_2_O_4_, and the corresponding FeO layer also becomes thinner. At this stage, the reaction rate of Fe–Cr alloy is still faster than that of pure iron. When the Cr content is gradually increased, (Fe, Cr)_2_O_4_ is formed, which in turn causes the parabolic rate constant to decrease [9]. However, since the rate of movement of Fe^3+^ is faster than that of Cr^3+^ ions, although the formation of an iron oxide layer with a mixed spinel structure reduces the oxidation rate, after a long period of oxidation, Fe oxide is still formed on the surface of the scale [10,11,12]. Only when the Cr content in Fe–Cr alloy is higher than 14%, a complete Cr_2_O_3_ layer can be formed on the surface of the alloy in the early stage of oxidation [13]. The formed complete Cr_2_O_3_ layer can prevent the outward diffusion of iron ions and inward diffusion of oxygen ions, thereby improving the anti-oxidation performance as well as reducing the oxidation rate of the alloy. If the concentration of Cr is lower than the critical concentration, a complete Cr_2_O_3_ layer cannot be formed, i.e., the stability of the protective oxidation product cannot be maintained for a long time [14,15]. Therefore, to ensure the heat resistance of Fe–Cr alloy, in most of the component systems, the Cr content exceeds 20% [16]. Previous studies have found that in the alloy-forming stable Cr_2_O_3_ layer, adding a small amount of Si can improve the ability of the alloy further to resist high-temperature corrosion [17,18]. In particular, Si can improve the reproduction of the Cr_2_O_3_ layer after the appearance of the surface oxide bubbles on the surface of the alloy under high temperatures [19,20].

In low-carbon steel, the content of Cr is often minimal (0%–1.15%), and the high-temperature oxidation behavior of this part of steel is entirely different from that of stainless steel [21]. The oxidation kinetics of Fe–Cr–Si ferrite steel is mostly dependent on the content of Si [22], and the results obtained from the studies of the effect of Cr content on the oxidation behavior of Fe–Cr–Si alloy are also not perfect. Based on this, in this study, the oxidation kinetics of Fe–1Cr–0.2Si alloy with low Cr content was first investigated, and then the high-temperature oxidation process of Fe–1Cr–0.2Si alloy was simulated under laboratory conditions. Then, the structure, thickness, and the mechanism of formation of the oxide were studied in combination with the results of oxidation kinetics.

## 2. Experimental

The research on the high-temperature oxidation behavior of Fe–1Cr–0.2Si steel is carried out through two experiments: oxidation kinetics and tube furnace oxidation. The elemental distribution of the test steel is shown in Table 1.

### 2.1. Oxidation Kinetics Experiment

To study the effect of Cr element on the oxidation kinetics of the tested steel, Fe–0.2Si steel was introduced as the steel for comparison. It does not contain Cr element, and the content of other elements remains the same as that of Fe–1Cr–0.2Si. Experimental steels were processed into 15 mm × 10 mm × 1.5 mm by electric discharge wire cutting, and 200#, 400#, 600#, 800#, 1000#, 1200#, and 1500# SiC sandpapers were used to grind the sample until the surface of the test piece was flat, and no apparent deviation in the thickness was noted. A drill bit with a diameter of 3 mm was used for drilling at the locations, as shown in Figure 1, and cleaned in a solution of acetone using an ultrasonic cleaner.

The treated samples were subjected to oxidation kinetics testing using a SETSYS ultra-high temperature synchronous thermal analyzer (SETARAM, Lyon, France). Argon gas was introduced into the temperature synchronous thermal analyzer at a rate of 20 mL/min, and the test piece was heated at a rate of 30 °C/min. When the set isothermal temperature (700 °C, 800 °C, 900 °C, 1000 °C, and 1100 °C) was reached, the system entered into the isothermal oxidation stage. At this time, the air was introduced into the furnace at a rate of 20 mL/min, where the sample enters into an oxidation process for 120 min. After the completion of the oxidation stage, the furnace was cooled to room temperature at a rate of 40 °C/min. During this process, the weight change of the test piece throughout the test was recorded using the electronic balance, which comes with the device.

### 2.2. Tube Furnace Oxidation Experiment

The tube furnace oxidation experiment was carried out based on the results of oxidation kinetics. In the oxidation kinetics experiment, Fe–1Cr–0.2Si steel was used. The experimental steel plate was cut into a test piece with a size of 30 mm × 26 mm × 3 mm using a wire cutter, and each specimen after cutting was ground with 200#, 400#, 800#, 1000#, 1200#, and 150# SiC sandpapers until the surface of the specimen was smooth, the direction of wear mark was the same, and there was no apparent deviation in the thickness. The test piece was washed with industrial ethanol for 5 min using an ultrasonic cleaner and finally blown using a hairdryer. In order to determine the formation mechanism of different oxidation products, the ion sputtering device was used to spray Pt on the local area of the sample surface before the tube furnace oxidation experiment. It is noted that the spraying time of Pt should be about 3 min. If the spraying time is too long, Pt forms a complete layer, which can block the diffusion reaction of metal cations and oxygen ions.

The treated samples were subjected to oxidation experiments in a tube furnace (SKGL-1200C, Shanghai Jujing Precision Instrument Manufacturing Co., Ltd. Shanghai, China). The tube furnace was heated from room temperature to the required isothermal temperature and then placed in a warm state. The porcelain boat with an experimental sample was placed in the middle of the quartz tube of the furnace. After 30 min, the sample in the porcelain boat was removed and placed in the air for cooling. The isothermal temperatures required for the experiments were 700 °C, 800 °C, 900 °C, 1000 °C, and 1100 °C.

After the experiment, the oxidized sample was processed into 10 mm × 6 mm × 3 mm by discharge wire cutting. The cut sample was inlaid using a ZXQ-2 metallographic inlay (WEIYI, Laizhou, Shandong). After the inlaid test piece, the surface of the inlaid sample to be observed was treated with SiC sandpaper of 800#, 1000#, 1200#, and 1500#, until the black solid oil stain and ripple caused by the wire cutting disappeared wholly. Then, the polishing process was carried out on the polishing machine with 2.5# and 1.5# diamond polishing paste. Finally, the surface of the polished test piece was quickly rinsed with alcohol, blown with cold air, and dried for testing.

The sectional morphology and elemental analysis of the scale were observed by the metallographic microscope, scanning electron microscope, and electron probe, and the phase of the scale was analyzed by XRD.

## 3. Results

### 3.1. Oxidation Kinetics

Figure 2 shows the macroscopic morphology of the scale produced using the TGA (Thermo Gravimetric Analyzer) test. It can be seen that the surface state of the oxidation product of each sample has a significant difference after being subjected to isothermal oxidation at different temperatures. A small quantity of iron oxide was formed at 700 °C, and a metallic luster was present on the surface at this temperature. When the oxidation temperature reached 800 °C or higher, the metallic luster disappeared, and the surface of the sample was covered with a layer of gray-black iron oxide. The surface of the scale formed by oxidation at 1100 °C has a distinct granular appearance, and the degree of oxidation is most severe.

The two materials were examined for the mass change, and the oxidation kinetic curve was plotted using Origin Lab 8, and the obtained results for Fe–1Cr–0.2Si steel and Fe–0.2Si steel are shown in Figure 3a,b, respectively.

It could be seen from Figure 3 that the slope of the weight gain curve of the two steel increases with an increase in the oxidation temperature. The oxidation weight gain per unit area of the material also increases with time at each temperature, and in the same steel, the oxidized weight per unit area increases with increasing temperature.

By comparing Figure 3a,b, however, it could be noticed that the increase in the rates of oxidative weights of the two steels is significantly different at the same oxidation temperature. Fe–1Cr–0.2Si steel has a lower rate of oxidation weight gain than Fe–0.2Si steel, and as the temperature increases, the time for linear oxidation decreases.

The oxidation weight gain per unit area of Fe–1Cr–0.2Si and Fe–0.2Si after oxidation for 120 min are shown in Figure 4. It could be observed that the oxidation weight gain per unit area of Fe–1Cr–0.2Si is always smaller than Fe–0.2Si steel.

### 3.2. Cross-Sectional Morphology of the Oxide Scale on Tube Furnace Oxidation

The cross-sectional morphology of the samples subjected to oxidation for 30 min from 700 to 1100 °C under a metallographic microscope is shown in Figure 5.

The overall state of the scale is in agreement with the image of the thermostatically oxidized surface. When the oxidation temperature is 700 °C, the surface of the test piece begins to have a small amount of incompletely distributed iron oxide scale. When the temperature is higher than 700 °C, the surface of the test piece shows a complete layer of iron oxide scale with a specific thickness and covers the substrate. Especially under the oxidation temperatures of 900 °C, 1000 °C, and 1100 °C, the iron oxide scale formed by the test piece showed an apparent layered structure. Therefore, the samples under these three temperatures were further examined by a metallographic microscope, scanning electron microscope, electron probe, and XRD. As shown in Figure 6, EDS analysis was carried out on the four typical positions of the sample at 900 °C, and the obtained composition results are shown in Table 2. The results of phase analysis of the scale using XRD performed at 900 °C are shown in Figure 7. A minimal amount of white Fe_2_O_3_ could be noted on the outermost side of the scale, and there is no formation of a continuous oxide layer. Below Fe_2_O_3_, there is a relatively thick layer of Fe_3_O_4._ The contrast of two layers (points 3 and 4) in Figure 6 is different, which displays a prominent two-layered structure. EDS and XRD analyses confirm that FeCr_2_O_4_ and Fe_2_SiO_4_ are contained within the two layers. In the layer, some Si and Cr elements and a relatively small quantity of Si and Cr elements are observed at the locations of point 3 and point 4, respectively. It is also possible for the formation of a composite oxide consisting of Fe, Si, Cr, and O at a position near the substrate side of the layer 4, but XRD could not detect these elements probably due to their small amounts.

As shown in Figure 8, EDS analysis was carried out on the four typical positions of the sample at 1000 °C, and the composition analysis results are shown in Table 3. XRD phase analysis was performed on the scale at this temperature, and the results are shown in Figure 9. From the combined analysis of EDS and XRD, the outermost layer is found to be the iron oxide layer, whereas the layer below it is a layer of Fe_3_O_4_. At 1000 °C, the FeO layer appears while there is a pro-eutectoid Fe_3_O_4_ in the FeO layer. FeCr_2_O_4_ also appears in the oxide layer near the side of the substrate. Although XRD did not detect Fe_2_SiO_4_, EDS analysis displayed the enrichment of Si and Cr elements near the substrate side. It is speculated that, on the one hand, due to high temperature and large thickness in the scale of the iron oxide layer, X-rays cannot be emitted to a position close to the side of the substrate, and hence the oxide of Si was not detected. On the other hand, it may be due to the low oxide content of silicon, which was not detected by XRD.

Figure 10 demonstrates the scanning electron micrograph of the sample section obtained at 1100 °C. Based on the metallographic images (Figure 5e), SEM, and XRD (Figure 11) analysis results, it can be seen that the outer to inner layers of the oxide iron sheet are, Fe_2_O_3_, Fe_3_O_4_, and FeO with pro-eutectoid organization, and the Si-rich and Cr-rich layers are close to the substrate side. Besides, the Si and Cr elements exist in the form of oxides of Fe_2_SiO_4_ and FeCr_2_O_4_.

Table 4 shows the results of surface scanning analysis of the secondary electron image of the iron oxide skin section and the corresponding Cr and Si elements under each experimental condition. At 900 °C, the enrichment of Cr and Si elements could be found on the side of the scale near the substrate. It is noted that the enrichment area is large, and the thickness is about 40–50 μm. The Cr and Si elements in the entire enrichment zone also do not have any apparent layered structure but are scattered throughout the enrichment zone. Combined with the analysis results of XRD as indicated in Figure 7, it can be proved that with oxidation at 900 °C, Cr and Si elements were mechanically mixed in the form of FeCr_2_O_4_ and Fe_2_SiO_4_, respectively, and there is no composite oxide phase consisting of Cr and Si elements. At 1000 °C, Cr- and Si-rich areas are found near the substrate side with a thickness of about 10–20 μm. As compared to 900 °C, the thickness of Cr- and Si-rich areas decreased, but the degree of enrichment increased significantly. The observations noted both at 1100 °C and 1000 °C are the same, except that the enrichment degree of Cr and Si was reduced at 1100 °C as compared to 1000 °C.

## 4. Discussion

### 4.1. Oxidation Kinetic Model

The alloy used in the experiment is continuously oxidized by contact with oxygen, and the generated oxidation product hinders the continuous movement of ions, thereby slowing down the oxidation [23,24,25]. The obtained oxidation curves conform to the parabolic law, i.e., the growth of the oxidation products conform to the parabolic equation. This could be expressed by the Kofstad equation [26], as shown below (Equation (1)):(1)$${\left(\Delta W\right)}^{2}=K_{p}t$$
where *ΔW* is the weight gain, *K_p_* is the oxidation rate constant, and *t* is the oxidation time. Table 5 shows the experimental *K_p_* values under various temperatures. The oxidation rate constant increases with an increase in the temperature, demonstrating that the amount of oxidation products obtained increases with an increase in the temperature.

According to Arrhenius and NevioBalo [27], the relationship between oxidation rate constant and activation energy of the steel can be expressed by the following Equation (2):(2)$$K_{p}=K_{0}\cdot \exp(-Q/RT)$$
where *K_0_* is the model constant, Q is the activation energy of the steel species (J/mol), T is the oxidation temperature (K), and R is the gas constant (8.314 J/(mol·K)).

Taking the logarithm, Equation (2) becomes as follows:(3)$$\ln K_{p}=\ln K_{0}+(-Q/RT)$$

Using the *K_p_*, *R*, and *T* values of the two steel grades at each experimental temperature, ln*K_p_* and *1/T* can be calculated. Based on Equation (3), a fitting line could be obtained by plotting ln*K_p_* vs. *1/T* at various temperatures, and the slope of the line is (−q/R). Then, the activation energy Q of the steel can be obtained.

The steel for this experiment was treated by the above method, and the obtained fitting curve is shown in Figure 12. The activation energies of Fe–1Cr–0.2Si and Fe–0.2Si steels were calculated to be 217.785 kJ/mol and 215.095 kJ/mol, respectively, and hence the activation energy of Fe–1Cr–0.2Si steel is slightly higher than the activation energy of Fe–0.2Si steel. This reveals that an increase in the Cr content slows down the oxidation reaction. Since the steel used in this research has less chromium content, its oxidation resistance is not significantly improved.

The linear law of high-temperature oxidation is affected by the conditions of external gas, and the oxidation rate increases with an increase in the flow rates of gas. Thus, oxidation was carried out under the above conditions. In the initial stage, since the iron oxide scale does not form completely, oxygen could sufficiently contact with the surface of the test piece, so that there was a linear relationship between weight gain and time due to oxidation. Specifically, it could be expressed by Equation (4):(4)$$W/A=k_{l}\times t$$
where *W* is the oxidation weight gain, *A* is the surface area of the test piece, *t* is the oxidation time, and *k_l_* is the linear reaction rate constant. The linearity data of the initial stage of the oxidation reaction of each test piece were fitted, and the obtained results are shown in Figure 13.

The fitting results of the linear oxidation rate constant obtained from the curve are shown in Table 6.

From Figure 14, it is observed that the time during which the entire oxidation phase exhibits linearity shortens with an increase in temperature. From Table 6, it can be noted that the values of the linear oxidation weight gain rate constant *k_l_* increase rapidly with an increase in temperature, and changes in the order of magnitude occur. These observations indicate that the temperature is a critical factor affecting the formation of oxidation products during the high-temperature oxidation of Fe–1Cr–0.2Si steel.

According to Figure 3 and Figure 5, the mass and thickness of the oxide scale are positively correlated with oxidation time. The oxidation process changes from a linear to parabolic pattern after reaching an absolute value. Chen and Yuen [28,29] calculated the quality of the scale at the moment of transformation by Equation (5):(5)$$W_{transition}=\frac{8f_{o}\rho k_{r}}{k_{l}}$$
where *W_Transition_* is the weight of the iron oxide scale when the linear law of the oxidation kinetics curve becomes a parabolic law, *K_r_* is the theoretical growth rate constant of FeO, *f_o_* is the mass fraction of oxygen in FeO, and *ρ* is the density of FeO at the oxidation temperature.

Equation (6) gives the relationship between *K_r_* and *K_p_*:(6)$$k_{p}=16fo\rho k_{r}$$

Rearranging Equations (5) and (6), Equation (7) could be obtained.
(7)$$W_{\mathrm{Transition}}=\frac{k_{p}}{2k_{l}}$$

Substituting *K_p_* and *K_l_*, the values of *W_Transition_* could be obtained at each temperature (Table 7).

It could be seen from Table 7 that the oxidation weight gain at the transition time increases at first and then decreases with an increase in temperature, reaching a maximum value of 5.39 × 10^−2^ mg·mm^−2^ at 900 °C. From this, it can be inferred that the oxidation law changes under 900 °C.

### 4.2. Oxidation Reaction

According to Ellingham’s diagram, by the relationship between the standard free energy of formation and temperature of oxides, the obstacles in the formation of different oxides under high-temperature oxidation can be judged [30]. When the temperature is above 900 °C, the partial pressure of oxygen in the air is higher than the partial equilibrium pressure of oxygen of SiO_2_, Cr_2_O_3_, Fe_2_O_3_, Fe_3_O_4_, and FeO in the initial stage of oxidation, and hence these oxides can be nucleated during this period. The degree of difficulty of their formation is in the following order: Fe_2_O_3_ > Fe_3_O_4_ > FeO > Cr_2_O_3_ > SiO_2_.

Based on the oxidation kinetics curve and cross-sections of the samples, the amount of iron oxide scale formed at 700 °C and 800 °C is relatively small. Although there is a distinct layered structure at 800 °C, Si was not detected by EDS, indicating that the diffusion rate of Cr element is higher than that of Si element in steel. All the four Fe, O, Cr, and Si elements are detected when the temperature is above 900 °C. The enrichment of elements concentrates gradually, and the layered oxide structure becomes more evident. At this temperature, the oxidative mechanism of alloy changes, which is consistent with the results of previous oxidation kinetics, and hence the oxidation process above 900 °C, is discussed in detail.

Considering Pt as the calibration material, the backscattered image of Fe–1Cr–0.2Si steel oxidized at 900 °C for 30 min, as shown in Figure 14. In this figure, the position of Pt is the original interface of Fe–1Cr–0.2Si steel [31]. The oxides of Fe nucleate preferentially on the surface of the alloy and grow outward at the same time. Most of the oxides of Fe are formed above the original interface, indicating that the principal reason for the oxidation of Fe is the outward diffusion of Fe ions [22], and the secondary reason is the inward diffusion of O ions. All the Cr and Si oxides are formed below the original interface, representing that the inward diffusion of O ions during the formation of Cr and Si oxides is the critical reason, and the outward diffusion of Fe ions is the secondary cause.

Since the inward diffusion of O ions predominates in the oxidation, Cr and Si ions preferentially form island-shaped Cr_2_O_3_ and SiO_2_ inside the alloy. However, since the amount of Fe ions in the alloy is much more significant than that of Cr and Si ions, the growth rate of Fe oxide is higher than that of Cr_2_O_3_ and SiO_2_ [32]. They nucleate and grow to form an outer oxide layer, which grows to the inside of the substrate; the preferentially formed FeO surrounds the island-like Cr_2_O_3_ and SiO_2_. Moreover, FeCr_2_O_4_ and Fe_2_SiO_4_ are formed by the solid-phase reaction (Equations (8) and (9)). When the FeO layer surrounds the island-shaped Cr_2_O_3_ and SiO_2_, not only the solid-phase reaction occurs but also the transformation from the internal oxidation to the outer oxidation, causing the original interface of the matrix to move up.
(8)$$FeO+Cr_{2}O_{3}=FeCr_{2}O_{4}$$
(9)$$2FeO+SiO_{2}=Fe_{2}SiO_{4}$$

The oxidation of the surface oxide layer of the alloy at three oxidation temperatures (900 °C, 1000 °C, and 1100 °C) is shown in Figure 15. At 900 °C, the diffusion coefficient of each ion is relatively small, so that the growth rate of FeO is quite low, and a part of FeO reacts with Cr_2_O_3_ and SiO_2_ to form FeCr_2_O_4_ and Fe_2_SiO_4_. When island-like FeCr_2_O_4_ and Fe_2_SiO_4_ are formed in the FeO layer, their spinel structure leads to a decrease in the diffusion rate of cations [33,34], which becomes an external diffusion barrier for the Fe ions. The FeO formed on the outer layer is oxidized with the external oxygen ions under the condition that the supply of Fe ions is insufficient, and Fe_3_O_4_ is formed, accompanied by a small amount of Fe_2_O_3_ nucleation. However, due to the lower oxidation temperature, the island-shaped FeCr_2_O_4_ and Fe_2_SiO_4_ formed at this time have a relatively small volume fraction in the FeO layer.

As the oxidation temperature increases, the diffusion coefficient of the outward diffusion of the cations and inward diffusion of the oxygen ions increases sharply [35,36], and the concentration of vacancies inside the matrix also increases [37]. The diffusion coefficient of Fe ions increases at 1000 °C. A thick FeO layer is formed when the island-like FeCr_2_O_4_ and Fe_2_SiO_4_ have not been formed, or the amount of formation is small. By extending the oxidation time, the formation of island-like FeCr_2_O_4_ and Fe_2_SiO_4_ increases, and the volume fraction of spinel increases. The density of the spinel in the FeO layer also increases, leading to an increase in the enrichment of Cr and Si at the interface between the scale and substrate (Table 4). Similarly, the spinel structure of FeCr_2_O_4_ and Fe_2_SiO_4_ hinders the outward diffusion of Fe ions, resulting in the formation of a Fe_3_O_4_ layer and even a Fe_2_O_3_ layer in the case of insufficient Fe and sufficient oxygen ions. The mechanism of oxidation at 1100 °C is substantially the same as that of 1000 °C.

It is worth noting that, in this study, the Cr content in the experimental material is 1%, and the oxidation product belongs to low Cr oxide, which cannot form a complete dense protective film. At the same time, since Cr cations are dissolved in the inner FeO layer and are in equilibrium with the second phase FeCr_2_O_4_, the oxide layer containing FeCr_2_O_4_ in the FeO layer has a distinct “doping effect” as compared to the pure Fe oxidation phase. This leads to an increase in the oxidation rate constant [38].

## 5. Conclusions

After 120 min of oxidation at the same temperature, the oxidation weight gain per unit area of the Fe–1Cr–0.2Si steel is more significant than that of Fe–0.2Si steel, and the activation energy of Fe–1Cr–0.2Si steel is slightly larger than that of Fe–0.2Si steel. Since the steel used in this research has less chromium content, its oxidation resistance is not significantly improved.The temperature is an essential factor affecting the formation of oxidation products in the high-temperature oxidation process of Fe–1Cr–0.2Si steel. The time for the linearity of the oxidation phase is shortened with an increase in the oxidation temperature. When the oxidation temperature exceeds 900 °C, the value of *W_Transition_* decreases, and the oxidation rule changes.When Fe–1Cr–0.2Si steel was oxidized at 900 °C, only a separate Fe_3_O_4_ layer in the iron oxide scale was noted, and no independent FeO layer was found out. Cr and Si elements enrich near the side of the substrate. At 1000 °C and 1100 °C, separate Fe_2_O_3_, Fe_3_O_4_, and FeO layers appear. The enrichment zone of Cr and Si also appears near the side of the substrate, and the degree of enrichment is more substantial than at 900 °C.The Cr and Si elements were enriched near the substrate side in the form of FeCr_2_O_4_ and Fe_2_SiO_4_, respectively, and no composite containing both Cr and Si was found.In the initial stage of oxidation, oxides such as SiO_2_, Cr_2_O_3_, Fe_2_O_3_, Fe_3_O_4_, and FeO were first formed. As the oxidation continued, the island-like Cr_2_O_3_ and SiO_2_ were surrounded by FeO, and then FeCr_2_O_4_ and Fe_2_SiO_4_ were formed by the solid-phase reaction, and the transition from internal oxidation to external oxidation was also completed.

## Figures and Tables

**Figure 1 materials-13-00509-f001:**
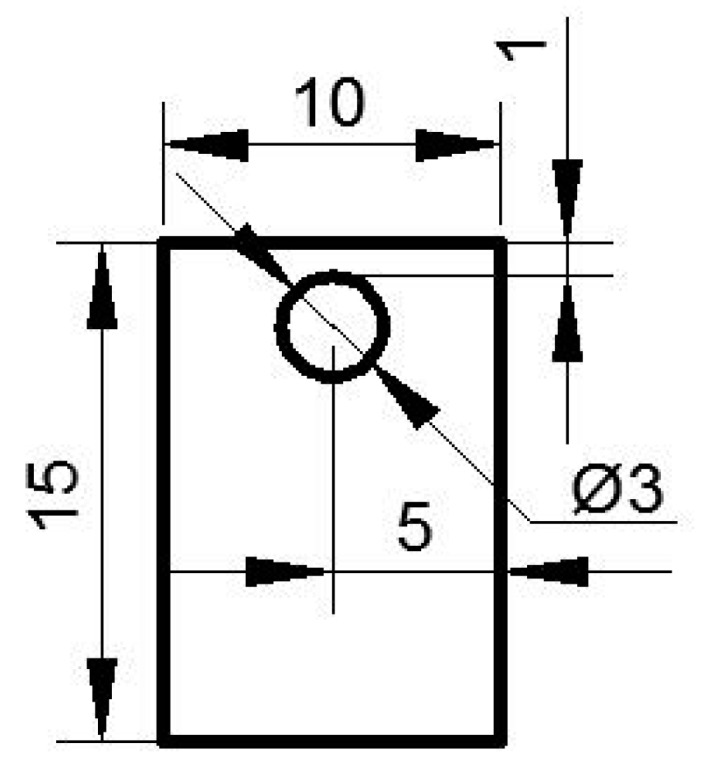
The location of the specimen borehole.

**Figure 2 materials-13-00509-f002:**
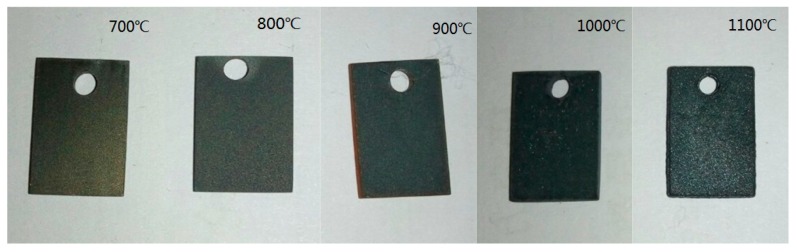
Macroscopic morphology of the Fe–1Cr–0.2Si steel specimens observed by the naked eye after 120 min of oxidation.

**Figure 3 materials-13-00509-f003:**
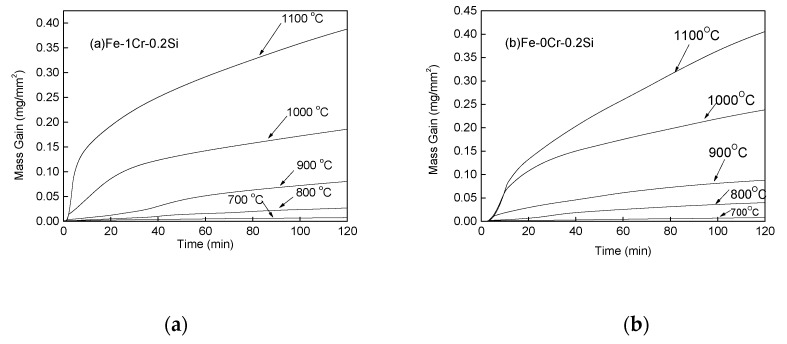
Weight gain curves of Fe–1Cr–0.2Si and Fe–0.2Si alloys exposed for 120 min. (**a**) Fe-1Cr-0.2Si; (**b**) Fe-0Cr-0.2Si.

**Figure 4 materials-13-00509-f004:**
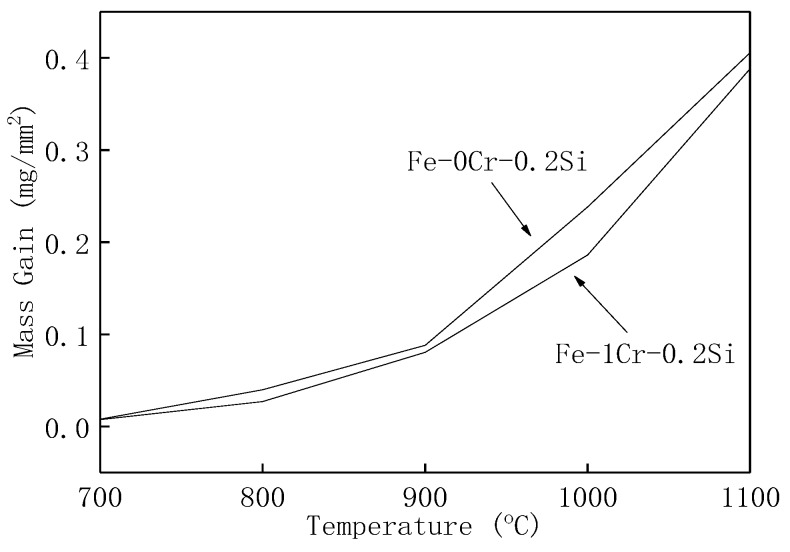
Increase in weight gain for two types of steel after 120 min of oxidation.

**Figure 5 materials-13-00509-f005:**
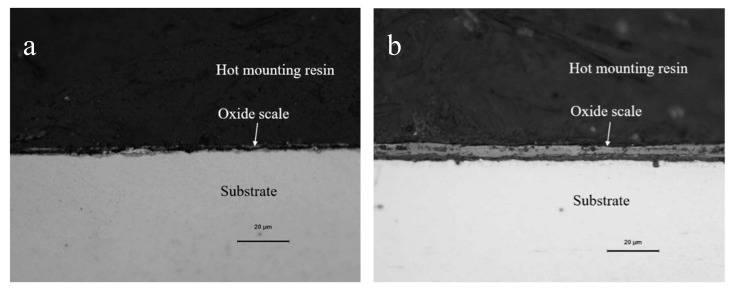
Cross-sectional morphology of the specimens observed under the metallurgical microscope. (**a**) 700 °C; (**b**) 800 °C; (**c**) 900 °C; (**d**) 1000 °C; (**e**) 1100 °C.

**Figure 6 materials-13-00509-f006:**
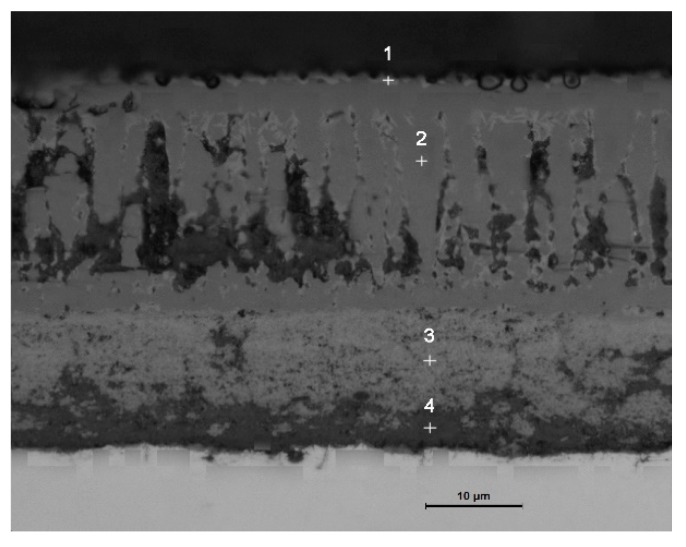
Cross-sectional morphology and EDS analysis at different positions of the iron oxide scale at 900 °C.

**Figure 7 materials-13-00509-f007:**
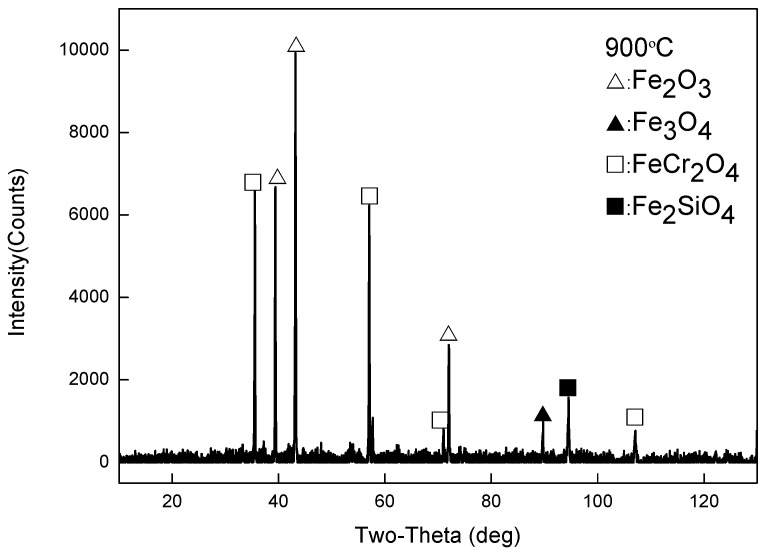
XRD data for the oxide scales obtained at 900 °C.

**Figure 8 materials-13-00509-f008:**
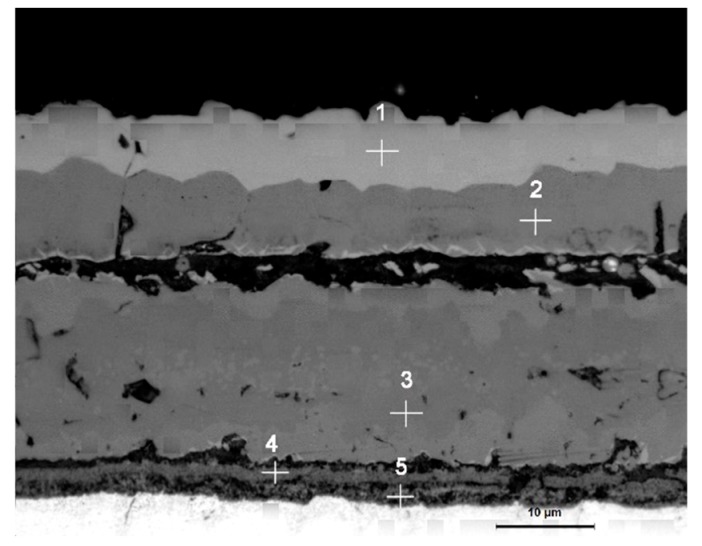
Cross-sectional morphology and EDS analysis at different positions of the iron oxide scale at 1000 °C.

**Figure 9 materials-13-00509-f009:**
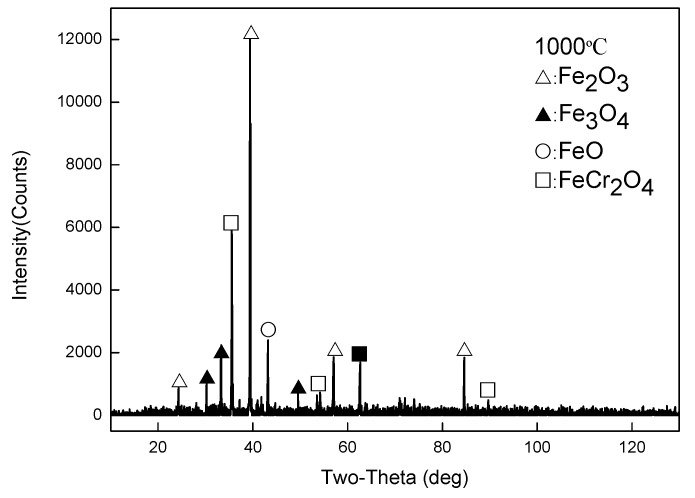
XRD data for the oxide scales obtained at 1000 °C.

**Figure 10 materials-13-00509-f010:**
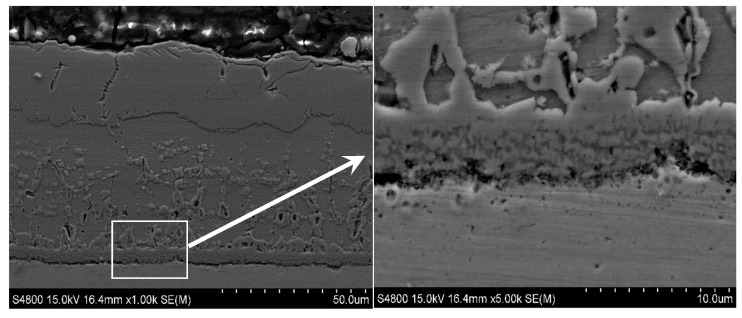
Cross-sectional image of the oxide scale obtained at 1100 °C.

**Figure 11 materials-13-00509-f011:**
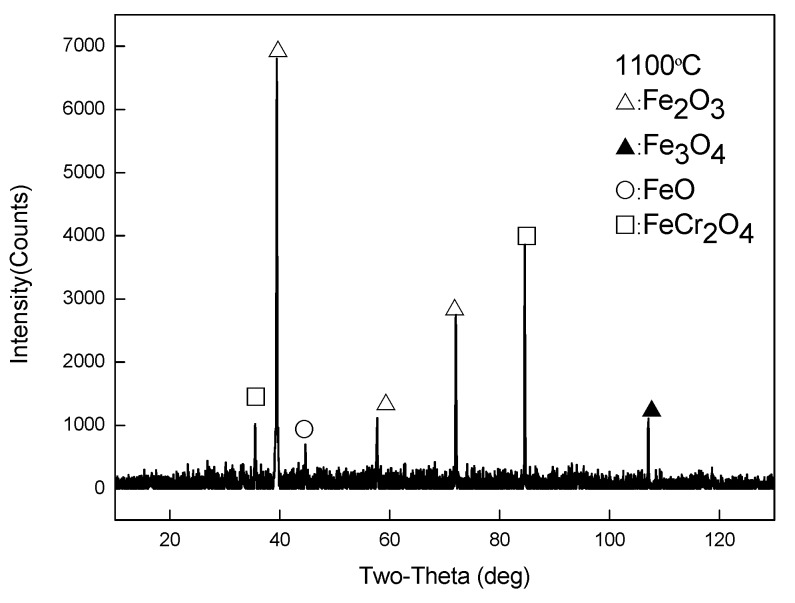
XRD data for the oxide scales obtained at 1000 °C.

**Figure 12 materials-13-00509-f012:**
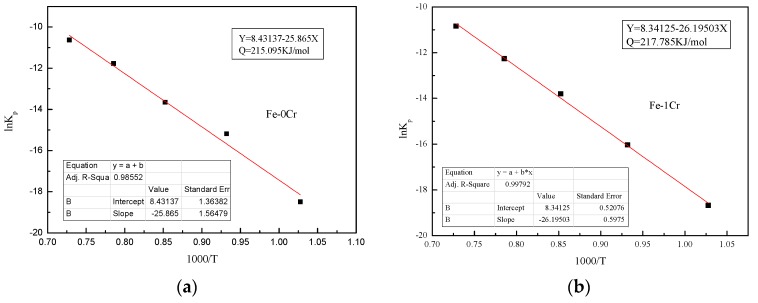
A fitting curve for steel (ln *Kp* vs. 1000/T). (**a**) Fe-1Cr; (**b**) Fe-0Cr.

**Figure 13 materials-13-00509-f013:**
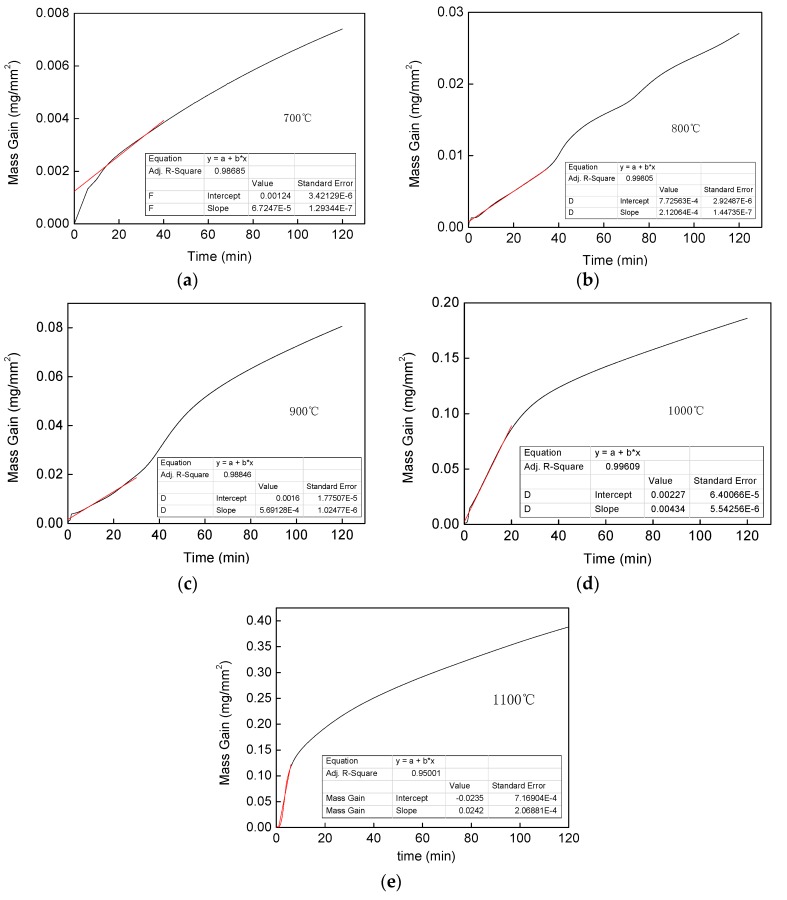
The kinetic plots obtained from the oxidation of steel in a short time. (**a**) 700 °C; (**b**) 800 °C; (**c**) 900 °C; (**d**) 1000 °C; (**e**) 1100 °C.

**Figure 14 materials-13-00509-f014:**
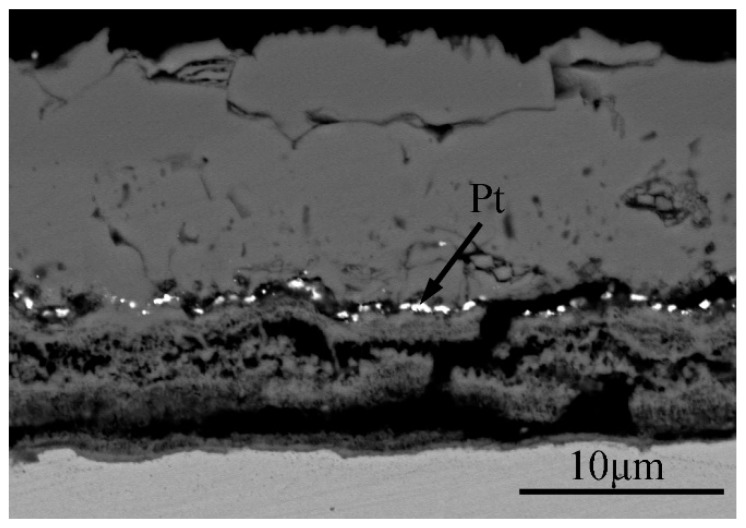
SEM backscattered electron image of the oxide scale obtained at 900 °C for 30 min.

**Figure 15 materials-13-00509-f015:**
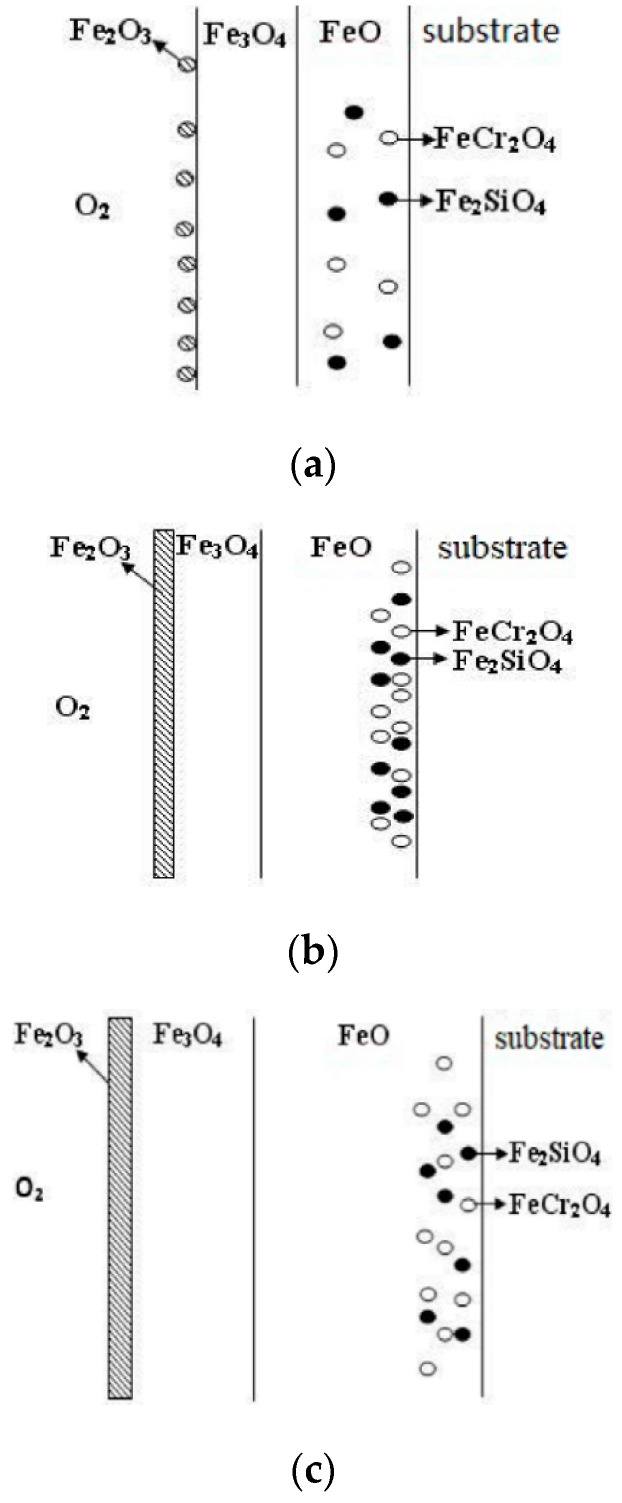
Schematic diagram of the cross-sectional micrograph obtained at different temperatures for the Fe–1Cr–0.2Si steel. (**a**) 900 °C; (**b**) 1000 °C; (**c**) 1100 °C.

**Table 1 materials-13-00509-t001:** Composition of the tested steels (%).

Sample	C	Cr	Si	P	Mn	S
Fe–1Cr–0.2Si	0.09	1.05	0.22	0.22	0.47	0.012

**Table 2 materials-13-00509-t002:** The elemental distribution of the energy spectrum.

Point Position	O	Fe	Si	Cr
1	58.26	41.74	-	-
2	56.69	43.31	-	-
3	55.13	35.89	3.42	5.57
4	60.33	26.40	5.12	8.14

**Table 3 materials-13-00509-t003:** The elemental distribution of the energy spectrum.

Point Position	O	Fe	Si	Cr
1	58.26	41.74	-	-
2	56.69	43.31	-	-
3	52.13	47.87	-	-
4	55.13	35.89	3.42	5.57
5	60.33	26.40	5.12	8.14

**Table 4 materials-13-00509-t004:** EDS elemental maps obtained from the cross-section of the Fe–1Cr–0.2Si steel oxidized at different temperatures.

Temperature	Secondary Electron Image	Cr	Si
900 °C	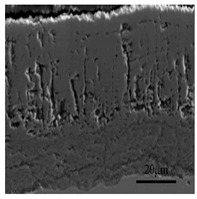	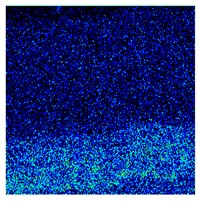	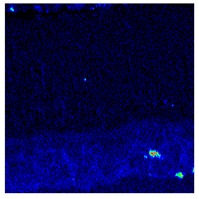
1000 °C	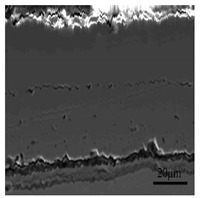	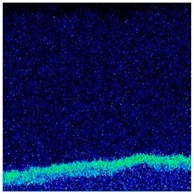	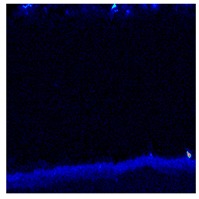
1100 °C	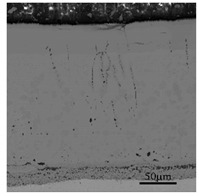	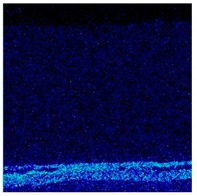	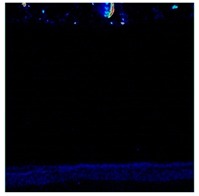

**Table 5 materials-13-00509-t005:** *K_p_* values of the alloy obtained at different temperatures.

Temperature (°C)	*K_p_* (mg^2^mm^−4^s^−1^)	
	Fe–1Cr–0.2Si	Fe–0.2Si
700	7.75 × 10^−9^	9.38 × 10^−9^
800	1.09 × 10^−7^	2.54 × 10^−7^
900	1.02 × 10^−6^	1.18 × 10^−6^
1000	4.72 × 10^−6^	7.69 × 10^−6^
1100	1.96 × 10^−5^	2.42 × 10^−5^

**Table 6 materials-13-00509-t006:** The calculated oxidation weight gain line rate constant of Fe–1Cr–0.2Si steel obtained at different temperatures.

Temperature (°C)	*k_l_* (mg.mm^−2^.min^−1^)
700	6.7247 × 10^−5^
800	2.12064 × 10^−4^
900	5.69128 × 10^−4^
1000	4.34 × 10^−3^
1100	24.2 × 10^−3^

**Table 7 materials-13-00509-t007:** The values of *W_Transition_* of the alloy at different temperatures.

Temperature (°C)	*W_Transition_* (mg·mm^−2^)
700	0.003457189
800	0.015404453
900	0.053911422
1000	0.032651959
1100	0.024373388

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
