# Peer review of "High-Temperature Oxidation Behavior of Fe–1Cr–0.2Si Steel"

_materials, 2020, doi:10.3390/ma13030509_

Round 1

Reviewer 1 Report

see attached file

Reviewer 2 Report

This paper presents investigation high-temperature oxidation behavior of Fe-1Cr-0.2Si steel. This steel oxidation were carried out in the tube furnace for 120 min and 30 min. Authors observed that with an increase in the oxidation time, the oxidative weight gain per unit area of Fe-1Cr-0.2Si steel changed from a linear to a parabolic relationship. The overall iron oxide structure of Fe-1Cr-0.2 Si steel is divided into two layers. The formation of the outer oxide of iron is mainly caused by the outward diffusion of cation, while the inward diffusion of O ion causes the formation of the inner oxide of chromium and silicon. As the temperature increases, the thickness of the outer iron oxide gradually increases, and the thickness ratio of the inner mixed layers of chromium and silicon rich oxides decreases.

In my opinion this paper can be interesting to readers of Materials journal. The paper is rather clearly presented. English of the paper is rather good and meet the requirement of the journal – in my opinion the language of the paper should be a little improved.

I find some editing mistakes for example:

There are often no spaces between words or sentences at work - e.g. lines 32 or 36 and many others. Most of figures are a good quality but figures 5, 6, 8 and Table 4 should be corrected for more readable. Please change a scale with resolution. Figure 15 is not a good quality – too small descriptions of relationships - not legible Amount of references is also sufficient but some papers cited in the references (26 from all 36) are older then 10 years and 33 from all 36 are older then 5 years. Minimum 23 references are from Asia (China, Japan or India). It would be desirable to expand this list somewhat by adding the work of other authors in the field of research over the past five years. You should past some references from America and Europe. I can recommend authors on this subject: A.D. Pogrebnjak or A.K. Fedotov. That can help to emphasize the relevance and significance of this study.

The manuscript can be accepted for publication after MINOR corrections.

Reviewer 3 Report

The paper deals with the influence of  1% Cr addition on the oxidation behaviour of Fe-0.2Si alloy for short exposure times (up to 120 min) between 700 and 1200°C. The authors reports that Cr addition reduces slightly the oxidation rate of the material. Major concern regards correlation between cross sectional observations and XRD measurements. There is no good agreement. In general the entire manuscript should be improved prior to accept it for publishing. Here the authors can found a list of points that should be addressed in a new version of the manuscript.

1) The paper compares the oxidation behaviour of two alloys for short exposures of 30 and 120 min. However, only characterization of the oxide scales formed in samples oxidized for 30 min is presented. No data (XRD or cross sectional views) regarding samples oxidized for 120 min is provided. This is neccesary because it could allow to clearly visualize how the oxidation progress during the steady state. Exposure after 30 min mainly provides information of the development of the scale during the transient stage . Moreover, no information is provided about the binary Fe-0.2Si alloy. 

2) There is no good agreement between the oxides identified by EDS micrioanalyses and XRD measurements. The authors states the formation of FeCr2O4 and Fe2SiO4,  but the stoichiometries of such phases, determined by EDS measurements are very far from the theoretical value. In addition, these phases appear at the innermost part of the scale. How peaks due to these phases are more intense thant that of Fe3O4 or FeO present in higher amounts in the scale. Moreover, why most intense peaks of the oxide formed at 900°C is Fe2O3 when the amount of this phase at the outermost surface is very low and much lower than that of Fe3O4?

3) Calculation of linear and parabolic oxidation constants have no sense in the present form. The authors should calculate the oxidation exponent to check the kinetics followed at each oxidation temperature. If the kinetics follow a parabolic behaviour the kp could be calculated and compared. Otherwise has no sense such comparison because the kinetics are different.

4) The authors consider always linear kinetics during the transient stage. This has no sense because is the time required until any scale begins to control the oxidation. Analysis made on this basis has no sense as well as data presented in Table 7.

5) The statement that Fe.02Si alloy is more likely to be oxidized than Fe-1Cr-0.2Si alloy based on activation energies is not true, especially considering the calculates values are essentially the same (218 and 215 kJ/mol).

6) What do you mean with "degree of difficulty". Are tou referring to kinetics or thermodynamic considerations?

7) Describe the use of Pt markers in the experimental section. 

Round 2

Reviewer 1 Report

I accept all corrections mde by the authors.